# How the Other Half Lives: What p53 Does When It Is Not Being a Transcription Factor

**DOI:** 10.3390/ijms21010013

**Published:** 2019-12-18

**Authors:** Teresa Ho, Ban Xiong Tan, David Lane

**Affiliations:** p53 Lab, Agency for Science, Technology and Research (A*STAR), Singapore 138648, Singapore; teresa_ho@p53lab.a-star.edu.sg (T.H.); bxtan@p53lab.a-star.edu.sg (B.X.T.)

**Keywords:** p53, transcription-independent, replication stress, DNA repair, apoptosis, centrosome, transposition

## Abstract

It has been four decades since the discovery of p53, the designated ‘Guardian of the Genome’. P53 is primarily known as a master transcription factor and critical tumor suppressor, with countless studies detailing the mechanisms by which it regulates a host of gene targets and their consequent signaling pathways. However, transcription-independent functions of p53 also strongly define its tumor-suppressive capabilities and recent findings shed light on the molecular mechanisms hinted at by earlier efforts. This review highlights the transcription-independent mechanisms by which p53 influences the cellular response to genomic instability (in the form of replication stress, centrosome homeostasis, and transposition) and cell death. We also pinpoint areas for further investigation in order to better understand the context dependency of p53 transcription-independent functions and how these are perturbed when *TP53* is mutated in human cancer.

## 1. Introduction

Since its discovery 40 years ago, p53 is first and foremost known as a master transcription factor and critical tumor suppressor. This still enigmatic protein is rich in functional domains, which work together in an intricate and, sometimes, convoluted manner to effect a myriad of functions. P53 contains two N-terminal transactivation domains (TAD1 and TAD2), a proline-rich domain, a core DNA-binding domain (DBD), an oligomerization domain (OD), and an unstructured C-terminal domain (CTD).

As a transcription factor, p53 assembles as a tetramer and the DBD recognizes sequence-specific response elements. The two functionally distinct TADs interact with a range of transcriptional cofactors that help dictate p53’s function in a context-dependent manner. The CTD, with its non-sequence specific DNA-binding ability, can reinforce the interactions between the core DBD and p53 target sequences.

Keen efforts have revealed that p53 is capable of recognizing and binding highly conserved response elements regardless of the chromatin landscape and compaction [1]. This interesting feature sets it apart from transcription factors that typically act in concert through the formation of complexes. Such a mechanism of action then raises the question of how the p53 transcriptional program is being regulated in a manner that allows for exquisite functions in different tissue environments, in development and disease, and in response to a variety of cellular stimuli and stresses.

To this end, studies have proposed the existence of factors (namely other members of the p53 family, such as p63 and p73) that can recruit chromatin remodelers that either modulate the immediate epigenetic landscape (i.e., methylation, acetylation) of the p53-bound enhancer region [2,3] or longer-range interactions associated with the overall chromatin topology [4]. Furthermore, the inherent nature of the p53 target promoter, as defined by core promoter elements, can influence the assembly of the preinitiation complex and RNA polymerase II (RNAPII) occupancy [5]. The degree of flux in such regulatory mechanisms and the variations in global chromatin features across cell types and physiological states can indeed expand p53 transcription-dependent functions multifold, consistent with its association with numerous signaling pathways and cellular processes.

## 2. Transcription Does Not Explain It All

The extensive characterization of p53 transcriptional functions in tumor suppression has unwittingly revealed the existence of p53 DBD and TAD mutants, which have selectively lost the ability to transcriptionally regulate the cell cycle and/or apoptosis but are associated with tumor suppression (elaborated below). Furthermore, mouse models deficient in key p53 target genes still exhibit better tumor suppressive ability than *Trp53* (murine p53 gene) null mice. The identification of alternate biological functions of p53 that are independent of its transcription and transactivation abilities and inexplicable on the basis of aforementioned regulatory diversity lends support. Such findings offer deeper insight into the ‘everyday’ roles of p53 under physiological conditions that are devoid of oncogenic or stress stimuli. These scenarios challenge the notion that p53 transcription-dependent functions alone are critical in tumor suppression and beg the questions of what are (1) the contribution of transcription-independent functions; (2) their interplay with transcription-dependent functions and other key oncogenic events; and (3) the context dependency of their functional manifestation.

One of the first indications that p53 tumor suppression is not exclusively linked to its transcriptional function was the observation that mice depleted of p21 [6,7] or PUMA [8] were not as prone to early tumor onset as mice deficient for p53. Even more significantly, the complete deletion of p21, NOXA, and PUMA, which led to deficiencies in cell cycle arrest, apoptosis, and senescence, still enabled suppression of T-cell lymphomas in mice compared to loss of p53 [9]. Mice lacking p53 cell cycle target genes, such as *Gadd45a*, *Ptprv*, and *PML*, fail to develop spontaneous tumors [10,11,12]. The characterization of ‘separation-of-function’ DBD mutants, such as R172P (corresponding to human R175P), which is defective in apoptosis but retains cell cycle arrest and senescence capabilities, revealed that these mutant mice exhibited less severe tumors compared to null mice [13,14]. TAD mutants (L25Q/W26S) deficient in cell cycle arrest and apoptosis (but not senescence) are capable of suppressing *KRAS* G12D-induced lung tumorigenesis [15]. Mice bearing acetylation-deficient p53 mutants deficient in cell cycle arrest, apoptosis, and senescence can also suppress tumor formation [16]. The triple acetylation mutant, K117R/K161R/K162R, actually retains the ability to regulate p53 target genes implicated in metabolism and the removal of reactive oxygen species (ROS).

Is it possible, then, that classical p53 transcriptional regulation of proliferation and apoptosis are not the sole limiting factors in tumor suppression, but rather, p53 transcription-dependent roles in metabolism and DNA repair are equally significant? To this end, recent findings have implicated p53 transcriptional regulation of DNA repair as an important aspect of its tumor suppressive function [17]. The knockdown of target genes *Mlh1*, *Msh2*, *Rnf144b*, *Cav1*, and *Ddit4* accelerated MYC-driven lymphoma development to a similar extent as p53 depletion. However, the contribution of RNF144b, CAV1, and DDIT4 to cell cycle dynamics cannot be fully excluded. It is also possible that given the complexity of p53 regulatory mechanisms, more context-dependent targets and effector pathways have yet to be identified.

However, it does good to recall that p53 is an immensely domain-rich protein, with individual domains possessing unique properties that contribute to the overall function of p53. These domains are also capable of diverse interactions with other proteins and modulate signal transduction. In particular, the TAD and CTD have emerged as multi-functional binding sites for an ever-expanding interactome. These domains are also subject to extensive post-translational modifications, which can regulate protein stability, turnover, and cellular localization and subsequently, protein–protein interactions [18,19]. These transcription-independent protein interactions are a significant part of p53 tumor-suppressive activity. In the following sections, we highlight the transcription-independent roles of p53 as they pertain to key biological functions in the cellular response to replication stress and DNA damage, apoptosis, centrosome integrity, and transposition.

## 3. Replication Stress Response

Several recent studies focused on the isolation of proteins on nascent DNA (iPOND) under normal physiological conditions and following replication stress have identified p53 as being a component of the replisome machinery at active forks. Proximity ligation-based assays have also revealed dynamic in situ interactions between p53 and components of the replication/transcription machineries. Such associations are facilitated by the ability of p53 to interact with replication protein A (RPA) [20] and the presence of sequence-specific (DBD) and non-sequence specific (CTD) domains within p53. The CTD has been reported to have a strong affinity for DNA with single-stranded gaps, triple-stranded DNA [21,22], and DNA duplexes with free ends or insertion-deletion lesion mismatches [23]. P53 can also catalyze DNA and RNA strand transfer and renaturation, promoting the annealing of complementary DNA and RNA single strands [24,25].

The aforementioned features would strongly support a direct transcription-independent role for p53 in replication. Furthermore, it appears that classical p53 transcriptional targets, such as p21 and Mdm2, are not induced following certain forms of replication blockade as compared to DNA-damaging conditions, such as ionizing radiation [26], suggesting that its transcriptional function is secondary with regards to specific insults.

A recent study by Hampp et al. highlighted the subtle differences in p53-mediated processing of stalled forks that promote either DNA resection or damage tolerance and bypass in order to maintain integrity [27]. During normal unperturbed replication, p53 interacts with the translesion polymerase, POLι, to decelerate nascent DNA elongation. Subsequently, a complex comprising of p53, helicase-like transcription factor (HLTF) and the SWI/SNF catalytic subunit (SNF2) translocase zinc finger ran-binding domain containing 3 (ZRANB3), promotes recombination and damage bypass. This concurs with other studies showing that p53 can enhance the replication fidelity of select DNA polymerases in vitro [28]. Meanwhile, in the presence of replication stress or persistent fork stalling, p53 and POLι promote MRE11-driven RPA accumulation on single-stranded DNA (ssDNA) to bring about more extensive remodeling and recombination of stalled forks. The above dual functions of p53 are dependent on its exonuclease activity rather than transcription capability. A p53 mutant, H115N, which lacks exonuclease activity and is transcriptionally more potent than wildtype p53 in inducing p21, is unable to process stalled forks [29].

Mechanistically, this study highlighted the subtle elegance of the p53-mediated response to low levels of endogenous replicative stress borne from normal replication processes (where its preferential to slow down replication to afford the cell more time to decide on the appropriate damage tolerance pathway) versus more persistent replication stalling (where forks might have permanently collapsed to form DSBs and would warrant more extensive remodeling and involvement of repair mechanisms). Another study has demonstrated the importance of p53 in safeguarding against topological conflicts between replication and transcription machinery [30], which is an endogenous source of DNA damage. However, whether this mechanism is dependent on or independent of p53 transcriptional activity requires further investigation.

These recently elucidated mechanisms above can also explain an earlier observation of p53 as having a transcription-independent suppressive effect on DNA synthesis in mouse zygotes fertilized by irradiated sperm [31], where the presence of DNA damage would have been a barrier to replication fork progression. These mechanisms also add much insight to earlier studies done in wildtype and mutant p53 backgrounds that suggest a seemingly transcription-independent role for p53 in replication dynamics.

A more detailed look into p53 function in the restart of stalled forks was reported by Roy et al. [32]. The investigators validated the presence of p53 at stalled forks using iPOND analysis. P53 promoted recruitment of MRE11 restart nuclease and MLL3, which brought about chromatin remodeling through histone H3K4 methylation. Interestingly, the study established that p53 was also critical in the suppression of mutagenic single-strand annealing (SSA) and microhomology-mediated end-joining (MMEJ) pathways mediated by RAD52 and POLθ, respectively. This significant role of p53 in ensuring that stalled forks are not hijacked by error-prone mechanisms is reinforced by an increase in spontaneous sister-chromatid exchanges under conditions of deficient or mutant p53 and the observation of mutational signatures consistent with POLθ-mediated mutations in p53-defective cancers [32].

Interestingly, the analysis of separation-of-function mutants revealed that p53 transcription-independent fork restart capability correlated better with tumor suppression. P47S mutant mouse embryonic fibroblasts (MEFs) exhibited increased fork stalling following treatment with hydroxyurea, comparable with p53-null MEFs. This is in spite of retention of transcriptional activity, as P47S is able to activate p21 and apoptosis. Mutant mice are tumor prone and P47S is a breast cancer predisposition polymorphism mainly in African populations [33,34]. Meanwhile, murine R172P (human R175P) is transcriptionally defective and yet exhibits improved fork restart capabilities in vitro. R172P mutant mice also retain partial tumor suppressive capabilities [13]. Hence, in order to maintain genome stability, p53 ideally functions to ensure continuity of replication, with minimal involvement of more mutagenic, damage-tolerant pathways.

Given that *TP53* is often mutated in a variety of cancers, how are these transcription-independent functions of p53 perturbed, especially in the case of ‘hotspot’ DBD mutants? The interaction between mutant p53 and MRE11 appears to be ‘preserved’ in the case of R273H and R248W mutants. In fact, this interaction disrupts MRE11 recruitment to ATM at double-strand breaks (DSBs), resulting in checkpoint attenuation [35]. Whether MRE11 recruitment to sites of stalled forks and its subsequent association with remodeling complexes as detailed in Roy at el. and the resultant effect on fork restart is relevant in the presence of mutant p53 is an area that will require further study. The same goes for the impact of mutant p53 on interactions with RAD52, POLθ and other translesion polymerases. In the presence of exogenous replication stress, the hotspot mutants R175H and R273H have also been shown to induce TopBP1 oligomerization, which attenuates its ability to activate ATR-mediated checkpoints that control origin firing and downstream phosphorylation of CHK1 [36].

We also need to probe these p53 mechanisms under unperturbed conditions and following exogenous stress in the form of conventional chemotherapy. The potential for mutant p53 to skew decisions between replication bypass, DNA resection, or fork restart mechanisms—all of which can, in turn, inadvertently boost cellular mutational burdens—has crucial implications for tumor development and drug resistance.

The new mechanistic evidences cited above strongly indicate that p53 affects critical transcription-independent roles in the maintenance of genome stability through the regulation of optimal rates of DNA elongation- and replication-associated recombination events. We need to better understand the sensitivity or thresholds of p53 transcription-independent function to different degrees of replicative stress and DNA damage. With improvements in high-resolution imaging and biochemical methods that allow for real-time assessment of the stability and affinity of protein interactions, we can gain insight into the domain-specific structural requirements of p53 function in modulating replication stress.

## 4. DNA Repair

The notion of transcription-independent functions of p53 in DNA repair had already been broached over two decades ago [22,23]. The DBD and CTD of p53 can recognize various aforementioned DNA damage-associated structures, and in vitro experiments revealed p53 binding to factors, such as TFIIH, RPA, DNA pol β, AP endonuclease (APE), Rad51, BLM, and WRN. P53 also possesses 3′ to 5′ exonuclease activity [28,37,38] and proofreading ability [39]. These properties were demonstrated in vitro, supporting the idea that p53 exonuclease function is not conferred by other proteins it may interact with, but rather is an intrinsic property of p53.

However, there is still a need for functional clarity of p53′s repair-associated interactions as evidenced from the studies cited below. Given the high degree of redundancy in repair mechanisms as revealed by CRISPR screens, understanding the kinetics and sequential order of p53 interactions with repair factors will be critical in determining how these functions influence repair and in what contexts.

Meanwhile, transcriptional targets of p53 pertaining to the different major repair pathways have received much attention. P53 regulates DDB2 [40] and XPC [41], which are involved in initial events of nucleotide excision repair (NER), OGG1 [42], and MUTYH [43] in base excision repair (BER) and MSH2, MLH1, and PMS2 in mismatch repair (MMR). P53 has been reported to interact with the RAD51 promoter, albeit with minor implications on its regulation [44]. As p53 has also been postulated to physically interact with these same transcriptional targets [45,46,47], it is crucial to understand the interplay between transcription-dependent and independent functions.

## 5. Homologous Recombination (HR)

The direct interactions between p53 and RAD51 and RAD54 have been relatively well characterized. P53 regulates RAD51 oligomerization [37] at sites of DSBs and blocks continued strand exchange, suppressing both spontaneous and damage-induced HR [48]. This is important in preserving genome stability by preventing excessive or inaccurate recombination events. This function of p53 was abolished by relevant RAD-binding mutants and dependent on p53 exonuclease activity, which is implicated in the destruction of heteroduplexes. Other studies have also reported that p53 interaction with DNA topoisomerase I [49] has a bearing on p53 transcription-independent regulation of HR. Mutants deficient in transactivation [50] (L22Q/W23S, 143V) and defective in DNA-binding and cell cycle regulation (138V) were still capable of full or partial repression of HR [51,52].

Replication stress can result in extensive ssDNA formation due to fork stalling or excessive origin firing. P53 suppression of HR of these stretches is dependent on its interaction with RPA [20] and phosphorylation by ATR [53]. Mutants that disrupt p53–RPA interaction (48H/49H and 53S/54S) but still retain transactivation and transcriptional activity are unable to suppress HR following exogenous replication stress or blockade [20]. Together, the above supports the notion that p53 has a direct role in HR stemming from its transcription-independent protein interactions, separate from transcriptional regulation.

Holliday junctions (HJs) are intermediates in HR and can also arise spontaneously during replication. The RecQ helicases, BLM and WRN, are responsible for unwinding HJs to reduce inappropriate recombination. P53 is capable of binding to BLM and WRN directly via its CTD, and attenuate their activity at HJs in vitro [54] and in vivo [55,56]. However, BLM and WRN are multifunctional proteins, and not only can they associate with a variety of nascent DNA strands and fork structures, they can also influence processivity and elongation of DNA replication [57]. In response to replication blockade by hydroxyurea (HU), phosphorylated p53 co-localizes with BLM and RAD51 at replication intermediates and this leads to HR suppression [58]. By doing so, p53 prevents the cell from processing stalled forks into HR substrates and allows other pathways (likely mediated by BLM and WRN) to restore forks. Most of these experiments comprise of in vitro binding assays and further in vivo verification is required for functional significance. Certain hotspot DBD mutants, R273H and R248W, are deficient in regulation of BLM and WRN and further work is required to determine the transcriptional independence of this particular p53 function.

Thus far, the studies above focus on HR in response to replication blockade and resultant intermediates, where p53 acts in a transcription-independent manner to suppress unscheduled or excessive HR. A recent study directly induced ‘clean’ DSBs at specific sites via the I-SceI endonuclease and reported that p53 regulation of HR was dependent on its transactivation function and coupled to p53 cell cycle regulation [59]. They also reported a slight increase in HR in p53-proficient cells in the presence of hotspot mutants, R175H and R273H. However, this increase in HR was mirrored by an increase in cells at the S and G2 phase, the cell cycle stages favorable for HR. The situation then seems to differ yet again in the case of replication-associated DSBs. Under these circumstances, p53 stimulates HR to prevent further fork collapse in a transcription-independent manner involving interactions with DNA topoisomerase I [52]. Yet, this stimulatory effect of p53 on HR in the case of DSBs is disputed by reports that suggest an inhibitory role of p53 [60], akin to that reviewed above for stalled replication. Such discrepancies can be attributed to the different types of agents used (i.e., HU, aphidicolin, doxorubicin, campthothecin, combination of drug treatment, and irradiation) and whether care was taken in determining the nature of the replication intermediates present and whether DSBs have fully formed from stalled or collapsed replication forks. While this pinpoints gaps in our understanding, it also highlights the subtleties of the p53 (transcriptional and non-transcriptional) function in maintaining the balance between the promotion and suppression of HR in response to different insults and the mechanisms by which they affect replication and subsequent DSB formation.

It is interesting that p53 retains its exonuclease activity when it is localized in the cytoplasm. In vitro biochemical analysis indicated that cytoplasmic exonuclease activity mirrored that of nuclear activity [61]. This is significant given that, under normal conditions, p53 is cytoplasmic during part of the cell cycle [62]. Breast and colorectal tumor cells also tend to sequester wildtype p53 in the cytoplasm [63,64,65]. Under normal conditions, does this retention of exonuclease activity potentially allow p53 to effect transcription-independent, recombination-mediated functions in relevant organelles with a genome-like mitochondria? In support of this, mitochondrial transcription factor A (TFAM) is known to interact with p53 and their interaction has been implicated in DNA damage sensing and repair [66]. DNA polymerase β and θ have been shown to interact with p53 to mediate repair and are implicated in mitochondrial DNA (mtDNA) repair [67]. Further investigation of relevant transcription-independent functions of p53 in mtDNA replication and repair is a worthwhile endeavor.

## 6. Pathway Choice: HR or Non-Homologous End Joining (NHEJ)

Unlike HR, the role of p53 in NHEJ is less well characterized. In general, a series of in vitro and in vivo studies have shown that p53 can rejoin or ligate DNA with DSBs [68,69]. Using the I-SceI endonuclease system, it was demonstrated that p53 has an inhibitory effect on error-prone or microhomology-mediated NHEJ (alt-NHEJ) [70]. This is in order to suppress genomic instability arising from low-fidelity repair. However, the study postulated that p53 might, conversely, promote error-free NHEJ (c-NHEJ), which involves simple rejoining of compatible ends.

The suggested mechanism by which p53 can promote NHEJ and/or dictate cellular choice between HR and NHEJ involves interactions between the DBD of p53 and the BRCT domain of 53BP1 [71]. In the absence of p53, 53BP1 foci formation at damage sites following ionizing radiation is impaired. Instead, there is an increase in BRCA1 recruitment, and these changes are independent of the degree of damage or cell cycle stage [72]. Such a transcription-independent role for p53 in defining the choice of repair pathway has significant implications for tumorigenesis and tumor response to chemotherapy. More in vivo work is required to elucidate the circumstances under which p53 drives repair choice, downstream factors involved in NHEJ that might also interact with p53, and how these mechanisms are affected in the presence of hotspot mutants. The latter point is interesting given that NHEJ efficiency has been reported to be more efficient than HR across all stages of the cell cycle [73], and this can have potential implications for mutational burdens and/or persistence of DNA damage in the absence of functional p53.

## 7. NER, BER, and MMR

NER is primarily responsible for the removal of helix-distorting lesions typically induced by UV irradiation. The role of p53 in promoting global genome NER (GG-NER) is more consistent across the literature compared to p53 function in transcription-coupled NER (TC-NER) as previously reviewed [46]. Firstly, p53 facilitates NER by promoting lesion recognition or detection. It does this by recruiting the p300 histone acetylase, which acetylases histone H3, leading to global chromatin relaxation and increased accessibility [74]. Subsequently, p53 binds and facilitates the recruitment of XPC, XPB, and CSB to photoproducts and cyclobutane pyrimidine dimers (CPDs) [75]. This is independent of p53 transcriptional activity. P53 can also modulate XPB and XPD helicase activity as demonstrated in vitro [76]. XPB and XPD are components of the 10-subunit TFIIH complex and p53 can influence the degree of DNA unwinding in the vicinity of a given lesion. However, more work is required to determine the effect of p53 on the overall efficiency of TFIIH, which also plays a role in transcription-coupled repair (TCR).

BER is the main repair choice for oxidative base modifications. Early studies have indicated that the presence of wildtype p53 elevates BER activity in vitro and that certain mutants deficient in transactivation or transcriptional ability are actually more effective [77]. This would suggest that either p53 transcribes targets that impeded BER or that these mutants may have enhanced the interactions or binding affinities with BER components. Subsequent work revealed that p53 can interact directly DNA polymerase β and stimulate BER in vitro and in vivo. The interaction was proposed to be mediated by TAD, as the transactivation defective p53 mutant, L22Q/W23S, failed to interact and was unable to stimulate BER [45,78].

MMR is mainly activated by the presence of erroneous nucleotides incorporated during replication. MSH2 is a major component of the MMR MSH2-MSH6 complex and is a known to be transcriptionally upregulated by p53 following UV [79]. Concurrently, p53 and MSH2 have been demonstrated to co-localize at early recombination intermediates [80,81], and depending on the cell cycle stage (i.e., S phase), can further bind RAD50 and RAD51 [82]. In vitro studies confirmed this interaction and revealed that it enhances binding of phosphorylated p53 (S392) to DNA with topological distortions [80,83]. While these p53-dependent mechanisms have been linked to MMR regulation, MSH2 has been implicated in a variety of repair pathways and it is necessary to determine if p53 function is pertinent and similar in these alternate pathways. An interesting notion is that, unlike the aforementioned repair pathways, p53 interacts with and transcriptionally regulates its gene target in MMR. More work is needed to determine if p53 transcription-dependent and independent functions work alongside in MMR or whether these functions are separate and dependent on the cellular insult or pathway choice.

The significance of p53 transcription-independent functions indeed cannot be understated. Several studies (and reviews) have suggested that the acute DNA damage response, which leads to p53 transcriptional activation of cell cycle arrest and/or apoptosis, is dispensable for tumor suppression [84,85,86]. However, these studies have mainly focused on p53′s major targets, p21, NOXA, and PUMA, in the context of acute DNA damage, and correlate hotspot and transactivation mutants, which are incapable of inducing cell cycle arrest and apoptosis, with a lack of tumor progression following acute damage. As these mutants might still retain some of the transcription-independent functions reviewed above, more detailed analysis is required. Tumorigenic cells also likely face lower levels of chronic or persistent DNA damage as opposed to the high levels induced by drugs or ionizing radiation, the latter of which are methods used to represent acute damage. Hence, in tumor cells, loss of p53 transcription-independent functions in damage sensing, repair choice, and fork processivity might have a greater overall impact on tumor progression.

Ultimately, as shown, transcription-independent functions of p53 are heavily dependent on the degree of DNA damage, cell cycle stage, and prevailing conditions, such as mutational burdens and the presence of other oncogenes in a given cell. Teasing apart and accurately modeling such subtle contextual differences in a physiologically relevant manner will be important in determining how exactly p53 dictates cell fate and outcomes.

## 8. Apoptosis

In mammalian cells, apoptosis is driven by two distinct pathways, the BCL-2 family-mediated mitochondrial pathway and the death receptor-mediated pathway, and converge onto a caspase-mediated pathway to elicit programmed cell death by the cleavage of several hundred cellular substrates, dismantling the cells from within. The two pathways can also collaborate, with the death receptor pathway triggering the BCL-2-regulated apoptotic programs.

The BCL-2 family members are related by virtue of the BCL-2 homology domains: BH1, BH2, BH3, and BH4, and can be subdivided based on their functions. The multi-domain anti-apoptotic members, including BCL-2, BCL-x_L_, and MCL-1, inhibit apoptosis by binding and sequestering the multi-domain pro-apoptotic members of the family BAX and BAK. Hence, the life of a cell hangs on an intricate equilibrium between these two branches of the family. The BH3-only proteins, such as BID, BAD, NOXA, and PUMA, act as the sensitizers and mediators, and tip the scale by disrupting the interaction between the multi-domain anti- and pro-apoptotic family members, or activating BAX and BAK to induce their oligomerization and the formation of pores in the mitochondrial outer membrane. Mitochondrial outer membrane permeabilization (MOMP) results in the release of cytochrome c, normally benignly involved in the electron transport chain for oxidative respiration but now drafted into programmed cell death. Cytosolic cytochrome c, together with APAF-1 and caspase-9, form the apoptosome, a complex that catalyzes the cleavage of procaspases into active caspases, bringing about the onset of apoptosis.

As a transcription factor, p53 directly upregulates the pro-apoptotic genes *NOXA* [87], *PUMA* [88,89], and indirectly *BIM* [90,91], and has been shown to inhibit the expression of BCL-2, either directly [92] or indirectly through miRNA34 upregulation [93]. P53 can also repress MCL-1 expression, though the mechanisms are less clear [94,95]. Although p53 can upregulate the expression of BAX [96,97] and APAF-1 [98,99], this upregulation does not appear to be essential, as p53 null hematopoietic cells express the same levels of BAX and APAF-1 as their wildtype counterparts, and similarly undergo apoptosis [100].

The observations that p53 translocates out of the nucleus upon cellular stress, and that apoptosis can occur in cells expressing transcriptionally dead p53 or dominant negative p53 provide evidence that there is more to p53 and apoptosis that can be attributed to its transcription activity. P53 has been reported to migrate to the mitochondria [101,102,103] and interact with members of the BCL-2 family by displacing anti-apoptotic members from pro-apoptotic BCL-2 proteins [104,105,106], or by directly activating BAX or BAK to induce MOMP [105,106,107]. P53 has even been reported to directly induce MOMP in ischemia models, independent of BAX/BAK but dependent on cyclophilin D [108]. Interestingly, many p53 mutants also lose the ability to induce apoptosis in a transcriptionally independent manner, despite retaining the ability to interact with BAK [105,107].

One criticism is that many of these earlier experiments were performed with anucleated cells, cell-free extracts, or isolated mitochondria, or that p53 was overexpressed ectopically beyond physiological levels. In those conditions, one may argue that using high concentrations of proteins, be they p53 or BAX, can lead to spontaneous aggregation and present observations akin to MOMP. For instance, in the seminal paper that provided evidence for p53 directly activating BAX, 100 nM of purified p53 was added to isolated mitochondria to induce cytochrome c release [106]. Similarly, 30 pmoL of purified GST-p53, or about 2 μg, was added to 30 μg of mitochondria to induce BAK oligomerization and cytochrome c release [104]. Non-physiological amounts of protein may be sufficient to perturb the delicate equilibrium preventing cells, already poised to commit apoptosis, to trigger irreversible cell death. In overexpression systems, confounding factors make it difficult to fully exclude the possibility that purportedly transcription-incompetent p53 mutants can still function as transcription factors at high enforced amounts.

With the advent of easily accessible gene editing, p53 can be mutated endogenously to avoid issues with overexpressing non-physiological levels of the potent tumor suppressor. Castrogiovanni et al. used CRISPR/Cas9 to introduce mutations to serine 392, a phosphorylation site that is important for mitochondrial translocation following genotoxic stress [109]. While the S392A phospho-incompetent mutant retains the ability to activate key p53 genes, including *p21*, *PUMA*, and *BAX*, it is unable to translocate the mitochondria following camptothecin treatment, and as a result the cells are deficient in inducing apoptosis compared to cells expressing wildtype p53. This suggests that p53 has a transcription-independent role at the mitochondria in inducing apoptosis on top of its function as a transcription factor, and the interplay of the two roles lower the threshold for bringing about cell death.

More evidence to suggest that p53 induces apoptosis independent of transcription can be seen with the R181E mutant [110]. This mutation abolishes the ability of p53 dimers to tetramerize onto DNA, rendering it incapable of transcription. Interestingly, unlike *Trp53* (murine p53 gene) knockout, the murine-equivalent R178E mutation is unable to rescue Mdm2-null mice from massive apoptosis that leads to embryonic lethality. Moreover, while Trp53R178E mutant mice phenocopy *Trp53*-null mice demonstrated extensive early onset tumors, Trp53R178E mutant mice responded favorably to chemotherapy unlike *Trp53*-null mice. P53 R178E was observed to translocate to the mitochondria in response to cytotoxic treatment, potentially mediating MOMP. Together, these data provide evidence to suggest that apoptosis can occur without p53-dependent transcription.

## 9. Centrosome Duplication

There has been a wealth of literature on the role of p53 in regulating the centrosome duplication cycle and its importance in safeguarding against abnormal amplification or reduplication and excess centrosomes, which lead to mitotic defects and chromosomal instability [111]. The significance of these findings is supported by the demonstration that centrosome amplification in the absence of p53 is a precursor to neoplasia, with implications for tumor initiation and progression [112].

This role of p53 is partially dependent on its transcriptional activation of p21. P21 regulates CDK2/cyclin E activation in order to coordinate initiation of centrosome and DNA replication. However, p21 is not solely responsible for p53-mediated function, as the introduction of p21 into p53-null cells only partially restores the centrosome profile while reintroduction of p53 fully recues it [113]. To this end, other p53 transcriptional targets, such as BubR1, which is implicated in centrosome homeostasis and number [114], may potentially be implicated.

Evidence for transcription-independent p53 functions in centrosome homeostasis stems from the ability of p53 to co-localize with centrosome proteins such as centrin, g-tubulin, and glutamylated tubulin [115], which may, in turn, confer a p53 ‘surveyor’ function. P53 co-localization is associated with inhibition of centrosome biogenesis while failure to interact is sufficient to initiate duplication. P53 localization also seems to be regulated by phosphorylation of the residue S315. A non-phosphorylatable mutant, S315A, localized only to duplicated centrosomes while a constitutively phosphorylated mutant, S315D, localized to duplicated and unduplicated centrosomes [116]. Both these mutants retain transactivation function but only S315D was capable of complete suppression of reduplication. Hence, it appears that suppression of reduplication was not entirely dependent on p53 transcriptional activation of p21 and that binding of p53 to unduplicated centrosomes was central to its regulatory functions.

Another study reported that p53 localization at centrosomes during mitosis and during the post-mitotic checkpoint was dependent on ATM phosphorylation of residue S15 [117], reinforcing a surveillance function for p53 in preventing centrosome reduplication following initiation of mitosis. This was also reported under conditions of mitotic stress, replicative or premature senescence or DNA damage where p53 centrosome localization is increased [118]. Following p53 localization under such conditions, 53BP1 and USP28 can mediate p53 activation and p21-dependent cell cycle arrest [119]. In addition, p53 and p38 have been shown to be co-recruited to centrosomes in response to a loss of centrosome integrity and this leads to inhibition of the G1-S transition [120,121].

Interestingly, investigation of transactivation-dead DBD mutants in a p53-null background revealed that different mutants can have distinct effects on centrosome homeostasis. Cells expressing the mutant R175H display exacerbated centrosome abnormalities compared to p53-null cells [111,116]. Meanwhile, R249S mutant cells exhibit significant suppression of centrosome reduplication compared to null and R175H conditions. The ability of these mutants to localize to centrosomes was accountable for these phenotypic differences given that both are transcriptionally deficient. Mutant R175H did not localize to unduplicated centrosomes while R249S, similar to wildtype p53, was detected at unduplicated and duplicated centrosomes. Another transactivation-dead mutant, D278N, which cannot activate p21 yet binds centrosomes, is capable of suppressing reduplication [122].

Ultimately, the studies above highlight a significant role for p53 in regulating centrosome homeostasis under normal cell cycle conditions and under stress. It is likely that both transcription-independent and -dependent functions of p53 work in concert to survey and regulate the centrosomal number and transduce relevant signals to the nucleus that result in transactivation of p21. To better separate these functions of p53, we need mechanistic insight into (1) how p53 recruitment is modulated under different physiological and stressed conditions, (2) how p53 association with key centrosome components enables inhibition of centriole elongation and reduplication, (3) high-resolution imaging of p53 at centrosomes in real-time during cell cycle progression, (4) the binding sites within p53, which contribute to its functions at centrosomes, and (5) the molecular events of signal transduction between the centrosome and nucleus and p53′s role in this.

Thus far, several molecular studies have been done using primary murine cells and have introduced human p53 hotspot mutants in p53-null mouse embryonic fibroblasts (MEFs). While it has been shown that essential centrosomal components are well conserved across diverse species [123], validation of the aforementioned results is necessary in human cells. This is especially so given reports where p53 loss in human cells did not result in as severe amplification and chromosomal instability as observed in murine backgrounds [124,125].

## 10. Suppression of Growth

Apart from the very well characterized p53-mediated growth arrest via its target p21, a transcription-independent process has been proposed. Gas1 is a plasma membrane protein highly expressed in G0. It can transduce anti-proliferative signals via the proline-rich domain (amino acids 63–85) of murine p53 [126] and hence link p53 with other signal transduction pathways [127]. Furthermore, the N-terminal transactivation domains of murine p53 are not essential for Gas1 growth suppression, as evidenced by TAD mutants that can still transmit Gas1 signals [128]. Corresponding investigation of human p53 and whether it affects similar functions as murine p53 will be important in determining if this transactivation-independent p53 function is conserved.

Another study also highlighted a role for cytoplasmic p53 in internalization of membrane receptors in order to regulate signal transduction. P53 can interact with clathrin-heavy chain (CHC) at the plasma membrane in order to regulate vesicle formation and endocytosis of epidermal growth factor receptor (EGFR) [129]. Hence, p53-mediated internalization and degradation of EGFR keeps a check on cell growth. In the presence of hotspot mutants R175H and R273H, increased recycling of EGFR and integrins by enhanced binding of RAB-coupling proteins to endosomes result in relocalization of EGFR to the plasma membrane and promotion of tumor growth and invasion [130]. Further investigation into this transcription-independent function of p53 in regulating other major growth factor receptors will prove insightful. This notion of p53 in transcription-independent signal transduction further expands our knowledge of p53 function in the cytoplasm and how this contributes to tumor suppression. Studying these effects in human cells under physiological/developmental and pathological states will also be important. In general, the cytoplasmic functions of p53 have been largely elucidated through in vitro studies and hence there is a need to evaluate these mechanisms in vivo to determine their impact on biological processes.

## 11. Suppression of Transposition

P53 has been implicated in the regulation of the movement and expression of transposons and other classes of repetitive elements in order to preserve genome integrity. A study by Wylie et al. [131] found that p53 interacts with piwi RNA (piRNA) protein complexes in female germ cells in *Drosophila* and zebrafish. Specifically, the presence of p53 limits transposition and results in epigenetic silencing of piRNA repeats [132]. In the absence of p53 or the presence of mutant p53, increased levels of mobile element RNA are observed and this correlates with increased occurrence of random integration events across the genome. The physiological impact is reduced fertility and defects in egg formation. The p53-mediated function of the suppression of P-element transposon movement has also been implicated in the maintenance of germline ovarian stem cells in *Drosophila* [133].

The mechanisms by which p53 suppresses transposition under physiological conditions appear to be independent of its transcriptional function. This is in spite of observations that p53-binding sites exist within various families of transposons, LINE elements [134], and long terminal repeat (LTR) retrotransposons [135]. Firstly, the aforementioned direct interaction between p53, piRNA, and associated complexes can lead to degradation of retrotransposon RNA. Secondly, negative regulation by p53 is associated with recruitment of chromatin remodelers, such as histone methyltransferase SET8 [136]. The role of p53 as a transcriptional repressor has been long debated [137]. However, increasing evidence has strengthened the notion that repression is an indirect effect of p53 that is either dependent on its downstream targets, like p21, Mdm2, and microRNAs, or a consequence of its ability to recruit and interact with repressors [1].

Meanwhile, under conditions of DNA DSBs, p53 transcriptional function comes into play at repetitive elements. p53 has been shown to be recruited to binding sites within transposons and repetitive elements to activate transcription and retrotransposition. This leads to increased insertion events and the resultant genomic instability creates a feedback loop, which, in turn, can reinforce cell death or senescence [132]. This truly demonstrates the context-specific or dual nature of p53 functioning at repetitive elements that engages both its transcription-dependent and independent capabilities. In a way, under normal conditions, p53 acts to suppress genomic instability contributed by transposition but enhances genomic instability by means of transposition following certain stresses or insults. The latter point is interesting given that some transposons are reportedly derepressed in human cancers [138] and the findings that common hotspot DBD mutants of p53 fail to suppress transposition [131].

Further work is necessary to (1) validate the evolutionary significance of the aforementioned mechanisms elucidated in *Drosophila* and zebrafish in human cells (a tantalizing observation being that human p53 can suppress transposition in *Drosophila*); (2) validate the existence of such mechanisms in preserving genome stability in somatic cells; (3) better characterize the dynamic protein interactor profiles of p53 and the mechanism of their recruitment to repetitive elements; (4) dissect the context dependency of p53 transcriptional and non-transcriptional functions; and (5) determine how these mechanisms are perturbed in the presence of mutant p53 and whether loss of repression is attributed to inability to bind to p53 response elements or changes in protein interactors.

## 12. A New Chapter Ahead

The increasing identification and characterization of non-transcriptional functions of p53 and their ultimate contribution to tumor suppression is shedding new light on the extensive capabilities of this multi-faceted protein (Figure 1). Moving forward, a more thorough understanding of how p53 transcription-independent functions are evoked in response to a variety of cellular insults is paramount. The structural requirements of p53 to effect transcription-independent functions (i.e., as a dimer or tetramer) will need to be elucidated. Investigating the subtleties in terms of thresholds of p53 function in different cellular and developmental contexts and interactions with other molecular pathways are equally important. Furthermore, to what extent are transcription-independent protein–protein interactions exacerbated or disrupted by mutant p53 and do these predominate transcription-dependent roles? What are the effects of structural changes in mutant p53 on the perturbation or preservation of transcription-independent functions? There must also be due considerations of the genetic background of the in vitro or in vivo models applied in future work. This concern is supported by numerous studies reporting contradictory results or enhanced or subdued effects when characterizing mutant p53 in homozygous mutant or p53 wildtype or null backgrounds. Despite it being four decades on from the initial discovery of p53, new and intriguing functions continue to be attributed to this critical guardian of the genome.

## Figures and Tables

**Figure 1 ijms-21-00013-f001:**
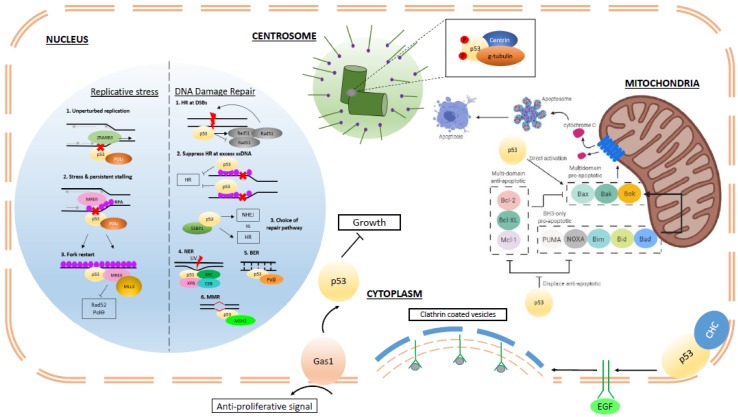
Summary of p53 transcription-independent functions. Within the nucleus, p53 regulates fork dynamics and processivity in response to endogenous and exogenous replicative stress through interaction with other key factors, such as MRE11, replication protein A (RPA), and translesion polymerases. (Grey arrows: direction of replication machinery; black arrows: direction of ZRANB3 translocase complex; Red cross: replication blockade; Red lightning: DNA damage (double or single-stranded) In the presence of damaged DNA, p53 regulates different repair mechanisms, such as homologous recombination (HR), by restricting excess recombination through interactions with Rad51 and RPA and nucleotide excision repair (NER), base excision repair (BER), and mismatch repair (MMR) through interactions with relevant components of the respective pathways as depicted. In the cytoplasm, p53 associates with centrosomal proteins, such as centrin and g-tubulin, in the regulation of centrosomal homeostasis and prevention of reduplication. (Red P: post-translational phosphorylation) P53 can transduce Gas1-mediated signals in order to regulate cell growth. Through its interaction with clathrin-heavy chains (CHC) at the plasma membrane, p53 can regulate endocytosis of EGFR and hence modulate the effects of growth factors on cellular growth and proliferation. Within the mitochondria, p53 can promote apoptosis through displacement of anti-apoptotic members of the BCL-2 family and from BCL-2 and directly activate BAX or BAK to induce mitochondrial outer membrane permeabilization (MOMP).

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
