# Peer review of "How the Other Half Lives: What p53 Does When It Is Not Being a Transcription Factor"

_ijms, 2019, doi:10.3390/ijms21010013_

Round 1
Reviewer 1 Report
The review paper submitted by Ho et al. aims to highlight the transcription-independent functions of the p53 protein. This quite comprehensive review is of great interest for those that want to know more about the star tumor suppressor protein. Indeed, the authors shed light in a precise and concise manner on the amazingly diverse effect of p53 beside its well know transcription factor function. The reader will thus be taken towards the replication stress response, DNA repair (including homologous recombination, non-homologous end joining, nucleotide excision repair, base excision repair, and mismatch repair), apoptosis, centrosome duplication, growth suppression, and finally suppression of transposition. At the end of the reading, the reader will be convinced of the multiplicity of p53's functions, well beyond its role as a transcription factor.
“Major” concern:
One could expect more than one Figure. Anyway, the unique Figure is to small and would benefit to be enlarged to a full-page landscape size.
Minor specific concerns:
- Trp53 (L 58, 397…) should be defined and explained.
- L 66-67, “One of the first indications …/… were the observations…”: “was” instead of “were”
- L 72, “(corresponding to human…”
- L 136, add a coma before “which”
Author Response
Point 1: One could expect more than one Figure. Anyway, the unique Figure is to small and would benefit to be enlarged to a full-page landscape size.
Response 1: We agree with the suggestion that the unique Figure should be enlarged to fill a page in landscape format. We believe that the Figure summarizes the key roles highlighted in our review.
Point 2: Trp53 (L 58, 397…) should be defined and explained
Response 2: We defined Trp53 as the murine gene equivalent of human TP53 in the text
Point 3: L 66-67, “One of the first indications …/… were the observations…”: “was” instead of “were”
Response 3: Corrected
Point 4: L 72, “(corresponding to human…”
Response 4: Corrected
Point 5: L 136, add a coma before “which”
Response 5: Corrected
Reviewer 2 Report
Overall this is a well written, concise and balanced review by a major leader in the p53 field. It integrates a long range memory of p53’s 40-year history of data collection with the newest findings on p53 transcription-independent functions, which represents an underemphasized facet of p53 function. This timely review is giving a critically important holistic view of the panoply of p53 functions in guarding genomic stability in health and disease by specifically regulating replication stress, apoptosis, centrosome duplication and transposition. The following points should be addressed before publication.
Major comments:
Lines 138-143: This small paragraph is unclear and a bit weak. Please sharpen its message. What exactly is “sperm-irradiated” ? Please clarify this unusual expression used by the original authors of Ref 31. Lines 141-143: add the corresponding Refs to the ‘earlier studies’ mentioned Lines 151-153: the text starting with “…is reinforced by…” should be deconvoluted and complemented by Refs 159-161: remind the reader here that the R172P mutant retains a partial tumor-suppressive phenotype Line 167: Is Ref (35) correct? Line 189: better “over 2 decades ago” Line 206: please add Refs to support the statement on physical interaction between p53 and these repair pathway proteins Line 259: Ref PMCID: PMC49686 should be added to Refs 60, 61 Line 284: what are knock-on effects Incomplete citations: Line 368: Ref 103 should be added to Refs 104, 105. Also in Line 370 on p53 mutants: Ref 103 (uses cells with endogenous p53 mutants) should be added to Ref 105. Moreover, the role of p53 in mediating necrosis and stroke induced by Ischemia-induced oxidative damage by regulating the mitochondrial permeability transition pore should be added (Cell. 2012 Jun 22;149(7):1536-48).
Minor comments:
Line 144, should say: “….into p53 function in restarting stalled forks…” Line 470: The correct Ref 126 citation is “Genes Cells. 2008 Apr;13(4):375-86” Line 293: typo, should read ‘dimers’ Line 489: please correct syntax
Author Response
Point 1: Lines 138-143: This small paragraph is unclear and a bit weak. Please sharpen its message. What exactly is “sperm-irradiated” ? Please clarify this unusual expression used by the original authors of Ref 31.
Response 1: We have clarified this in the text.
Point 2: Lines 141-143: add the corresponding Refs to the ‘earlier studies’ mentioned
Response 2: We have added the appropriate reference.
Point 3: Lines 151-153: the text starting with “…is reinforced by…” should be deconvoluted and complemented by Refs
Response 3: We have clarified this in the text.
Point 4: 159-161: remind the reader here that the R172P mutant retains a partial tumor-suppressive phenotype
Response 4: We have clarified this in the text.
Point 5: Line 167: Is Ref (35) correct?
Response 5: We have revised the reference, thank you!
Point 6: Line 189: better “over 2 decades ago”
Response 6: We have revised the text.
Point 7: Line 206: please add Refs to support the statement on physical interaction between p53 and these repair pathway proteins
Response 7: We have included appropriate references.
Point 8: Line 259: Ref PMCID: PMC49686 should be added to Refs 60, 61
Response 8: We have included the suggested references.
Point 9: Line 284: what are knock-on effects
Response 9: We have clarified this in the text.
Point 10: Incomplete citations: Line 368: Ref 103 should be added to Refs 104, 105. Also in Line 370 on p53 mutants: Ref 103 (uses cells with endogenous p53 mutants) should be added to Ref 105. Moreover, the role of p53 in mediating necrosis and stroke induced by Ischemia-induced oxidative damage by regulating the mitochondrial permeability transition pore should be added (Cell. 2012 Jun 22;149(7):1536-48).
Response 10: We have included the suggested references.
Point 11: Line 144, should say: “….into p53 function in restarting stalled forks…” Line 470: The correct Ref 126 citation is “Genes Cells. 2008 Apr;13(4):375-86” Line 293: typo, should read ‘dimers’ Line 489: please correct syntax
Response 11: We have revised the text accordingly.
Reviewer 3 Report
The review article highlights overlooked functions of p53 independent of its well studied transcription factor roles. This review does a good job at reviewing the "everyday roles" of p53 when cell stress is not ongoing as well as reviewing the sometimes perplexing tumorigenesis of transcriptional targets of p53 -null mouse models (i.e. p21-null, PUMA-null). The review explains these perplexing results described above by reviewing p53's role in direct regulation of DNA strand abnormalities, endonuclease activity, mitochondrial localization, centrosome recruitment and influence on retrotransposition. More importantly, the review identifies and speculates on the additional research needed to further advance the field of p53's transcriptional-independent roles in the cell and how this role influences homeostasis.
Some minor suggestions and edits include:
1 Early on on lines 55-56, it is mentioned that p53 mutants that "have selectively lost the ability to transcriptionally regulate the cell cycle and/or apoptosis..." it would be helpful if the authors gave a brief example as to how or what mutation(s) of p53 lends to a loss in transcriptional regulation of cell cycle (i.e. highlight mutations in the DBD of p53).
2 Line 312, what phosphorylation site(s) of p53 lend it to bind to distorted DNA topologies?
3 Line 380-382, p53 mutants that are overexpressed can be assessed for their abilities to be transcriptionally active possibly by assessing each mutant's binding affinity to target DNA sequences with techniques that can measure such interactions.
4 One subject matter not discussed in the review but may be of interest is viral inhibition of p53. It is known that viral proteins can block the transcriptional activity of p53 while still maintain p53 protein levels in the cell. What roles does p53-viral protein interactants have on p53's transcriptionally independent functions for cell homeostasis?
5 A table that can organize p53 mutations, their disruption of transcriptional activity of p53 and the mutants ability to maintain normal cell homeostasis would be very helpful summary to this review although not necessary.
Author Response
Point 1: Early on on lines 55-56, it is mentioned that p53 mutants that "have selectively lost the ability to transcriptionally regulate the cell cycle and/or apoptosis..." it would be helpful if the authors gave a brief example as to how or what mutation(s) of p53 lends to a loss in transcriptional regulation of cell cycle (i.e. highlight mutations in the DBD of p53).
Response 1: We have clarified this in the text.
Point 2: Line 312, what phosphorylation site(s) of p53 lend it to bind to distorted DNA topologies?
Response 2: We have included information on the phosphorylation sites.
Point 3: Line 380-382, p53 mutants that are overexpressed can be assessed for their abilities to be transcriptionally active possibly by assessing each mutant's binding affinity to target DNA sequences with techniques that can measure such interactions.
Response 3: We acknowledge that there are assays that can measure interactions between p53 mutants and target DNA, but there is a margin of error associated with ChIP-type experiments. We have revised our wording of the sentence accordingly.
Point 4: One subject matter not discussed in the review but may be of interest is viral inhibition of p53. It is known that viral proteins can block the transcriptional activity of p53 while still maintain p53 protein levels in the cell. What roles does p53-viral protein interactants have on p53's transcriptionally independent functions for cell homeostasis?
Response 4: This is indeed an interesting concept. However, we believe that the current literature puts this outside the scope of our review.
Reviewer 4 Report
The roles of p53 as a transcription factor is well known, but its other functions are still remained to be elucidated. In this review article, Ho and Lane et al comprehensively summarized the current findings of p53 function other than a transcription factor. This paper cover more than 100 literatures and discuss its functions in 9 categories in systematic ways. I recommend this paper to be published in the current form in International Journal of Molecular Sciences.
Author Response
We thank the reviewer for this comments.